# Assessing Long-Term Impact of Dietary Interventions on Occurrence of Symptoms Consistent with Hypoglycemia in Patients without Diabetes: A One-Year Follow-Up Study

**DOI:** 10.3390/nu14030497

**Published:** 2022-01-24

**Authors:** Marianna Hall, Magdalena Walicka, Mariusz Panczyk, Iwona Traczyk

**Affiliations:** 1Department of Human Nutrition, Faculty of Health Sciences, Medical University of Warsaw, 01-445 Warsaw, Poland; iwona.traczyk@wum.edu.pl; 2Department of Internal Diseases, Endocrinology and Diabetology, Central Clinical Hospital of the Ministry of Internal Affairs and Administration in Warsaw, 02-507 Warsaw, Poland; magdalena.walicka@cskmswia.gov.pl; 3Department of Human Epigenetics, Mossakowski Medical Research Institute Polish Academy of Sciences, 02-106 Warsaw, Poland; 4Department of Education and Research in Health Sciences, Faculty of Health Sciences, Medical University of Warsaw, 02-091 Warsaw, Poland; mariusz.panczyk@wum.edu.pl

**Keywords:** reactive hypoglycemia, dietary intervention, low-glycemic index diet, Mediterranean diet

## Abstract

Background: A well-balanced nutritional diet pattern has a significant role in the management of diet-related disorders. Currently, there are no specific dietary guidelines to refer to when advising non-diabetic patients with symptoms attributed to hypoglycemia in the postprandial period or patients with confirmed reactive hypoglycemia (RH). The aim of this study was to investigate the impact of the dietary interventions, and their sustained outcome, on the severity of hypoglycemic-like symptoms occurring in non-diabetic patients. Methods: The study group included forty non-diabetic individuals with symptoms consistent with RH. At the baseline, each patient underwent RH diagnosis and complex dietary evaluation. Over a period of six months, each patient had four appointments with a dietitian. Two sessions were focused on a dietary education about low glycemic index diet (LGID) and Mediterranean diet (MD). The said diets were to be followed for a period of three months, with two additional dietary check-ups. Once dietary supervision was completed, patients had no imposed dietary patterns. The final follow-up appointment took place twelve months later and that is when each patient underwent a detailed assessment of their current dietary habits and evaluation of the frequency of symptoms consistent with hypoglycemia. Results: There was a statistically significant reduction in the severity in eight out of the ten analyzed hypoglycemic-like symptoms after the dietary interventions. The most significant change was observed in the following symptoms: hunger (η^2^ = 0.66), impaired concentration (η^2^ = 0.61), hand tremor (η^2^ = 0.55), and fatigue (η^2^ = 0.51). The outcomes were comparable for both recommended diets, the LGID and the MD. The reduction in hypoglycemic-like symptoms continued after the twelve-month period. The individualized dietary counselling significantly improved the patients’ eating habits in comparison to those present prior to intervention in terms of healthy diet index (F_(2,78)_ = 27.30, *p* < 0.001, η^2^ = 0.41, 90%CI [0.26; 0.51]) and unhealthy diet index (F_(2,78)_ = 433.39, *p* < 0.001, η^2^ = 0.91, 90%CI [0.89; 0.93]). Conclusions: Healthy modifications in dietary habits may improve patient’s well-being and reduce the severity of their postprandial symptoms attributed to hypoglycemia. Therefore, dietary intervention focused on appropriate nutritional management combined with follow-up consultations may be a beneficial step towards comprehensive treatment of non-diabetic patients who present with hypoglycemic-like symptoms in the postprandial period.

## 1. Introduction

Hypoglycemia is a well-documented medical condition that mostly occurs in diabetic patients [1]. It appears when a glucose level falls below a defined physiological threshold and if further accompanied by adrenergic and neuroglycopenic symptoms.

The endocrine factors influencing glucose fluctuation have been established as insulin and glucagon. These two antagonistic hormones, which are produced by cells in the islets of Langerhans of the pancreas, are primarily responsible for blood glucose modulation, and their actions are widely determined by glucose fluctuation [2]. Moreover, there are well-identified variables influencing glycemic response in the postprandial and nocturnal periods. The most significant ones include physical activity, stress, infections, illnesses, and medication. In addition, the quantity and composition of the consumed meals, as well as fasting length, can affect glucose levels [3]. It should be noted that blood glucose concentrations fluctuate throughout the day. These physiological changes (if within the normal glucose level range) should not cause any alarming symptoms.

It should be emphasized that non-diabetic patients, without any insulin disturbance (insufficient secretion of insulin or impaired insulin action), may also experience some glycemic fluctuation with coexisting hypoglycemic-like symptoms. Reactive hypoglycemia is one of the examples of such an occurrence. RH can be described as postprandial hypoglycemia, which occurs 2–5 h after food intake. Although this condition has been identified in the early 20th century, until now there has been lack of its complete etiology and pathogenesis [4]. It is not uncommon that the patients who present with symptoms attributed to hypoglycemia have no glucose decline [5]. Their symptoms may occur in the postprandial period, but the frequency of those warning signs does not have a fully interpretable pattern. Due to the fact that hypoglycemic-like symptoms are not always associated with low glycemia, it is even more unclear how to interpret the occurrence of the abnormal symptoms attributed to hypoglycemia [6]. There is no standardized diagnostic procedure to determine the factors that contribute to and/or cause the symptoms in the postprandial period. Moreover, there are currently no guidelines relating to appropriate intervention in order to reduce hypoglycemic-like symptoms.

There have been many studies focusing on appropriate dietary interventions that can be helpful in reducing the possibility of unexpected glycemic fluctuation in patients with diabetes [7,8,9,10]. However, RH has been linked to subjects without metabolic disturbances, and there are no clear health-related factors that could be causing patients’ hypoglycemic symptoms, both at the onset of real hypoglycemia and when the blood glucose is within the normal range. Hypothetically, those disturbances could be triggered by factors associated with the patients’ lifestyle and diet pattern [4]. It has been widely acknowledged that diet modifications have a significant role in the treatment and management of diet-related diseases [11,12,13]. At present, there are no formal recommendations for patients without diabetes who experience symptoms consistent with hypoglycemia, which would outline the most effective nutritional pattern that should be followed. However, there are two diets that deserve particular attention in terms of the possible impact on reduction of glycemic fluctuation, which could be suitable for patients with reactive hypoglycemia. These are the low glycemic index diet and the Mediterranean diet [14,15].

The LGID is commonly described as a diet that can potentially reduce glucose surge in the postprandial period. Therefore, it is one of the recommended dietary patterns for people with diabetes and obesity [16,17,18]. The glycemic index is a parameter that reflects the increase in blood glucose concentration after the intake of certain types of food, compared to the increase in blood glucose concentration after the consumption of the same amount of carbohydrates in the form of pure glucose. Food can be classified as having a low, medium, or high glycemic index. Beneficial effects can be observed when patients are following a diet that contains products with a low glycemic index, as these are more likely to slow down the increase of blood glucose levels compared to foods with a medium or high index. Such a diet involves exclusion of certain vegetables, fruits, and grain products, as well as processed foods containing refined starches and added sugars [19]. In order to prepare the meals in line with LGID, it is necessary to use the appropriate tables, which set out the glycemic index values. However, there are some limitations in terms of the glycemic index. The specific product values may vary between tables; the index value does not correlate with the quantity of a consumed product; and those values can be further affected by food preparation method and meal composition [20].

A Mediterranean diet has also been proven to have beneficial, therapeutic effects on conditions such as obesity, metabolic syndrome (MetS), cardiovascular disease (CVD), and type 2 diabetes (T2DM) [21,22]. The MD is based on consumption of vegetables and whole grain products (which are rich in fiber), unsaturated fats (like olive oil, nuts, seeds), and protein rich food (lean meat, fish, eggs, legumes, and dairy products). These dietary components provide a favorable ratio of omega-6 and omega-3 essential fatty acids, and high intake of fiber, antioxidants, and polyphenols. The key element of this diet is its simplicity; there are no specific restrictions imposed when it comes to choosing the healthy, unprocessed products, which gives more options from a wide selection of the plant-based foods. MD is more likely to be categorized as an eating pattern rather than a structured diet [23].

The dietary pattern, in terms of meal composition and a frequency of consumption, can have a significant impact on the postprandial glycemic fluctuation. However, it remains unclear why certain non-diabetic patients experience symptoms attributed to hypoglycemia (with and without glucose decline), which factors increase the severity of these symptoms, and what may reduce the frequency of symptoms. There are currently no studies that analyze eating habits of individuals without metabolic disturbance and who manifest symptoms attributed to hypoglycemia in the postprandial period. Moreover, there is no research data available that provide a realistic reflection of the effect of dietary interventions and evaluation of patient adherence from the long-term perspective. Thus, the purpose of this preliminary study was to evaluate the long-term effect of the healthy dietary interventions on the occurrence of hypoglycemic-like symptoms in the postprandial period. The assessment also included the evaluation of eating habits and coexisting symptoms observed in the pre and post-intervention periods as well as the subsequent twelve-month period without any dietary supervision.

## 2. Materials and Methods

### 2.1. Participants

The study group included forty non-diabetic patients with symptoms attributed to hypoglycemia. The subjects were referred to the hospital in order to perform a detailed diagnosis toward suspected reactive hypoglycemia. Subjects underwent diagnostic procedures to screen for RH and other metabolic disturbances, which have been described in detail in the previous study [6]. The diagnostic tool to confirm RH was a five-hour oral glucose tolerance test that required administering orally seventy-five grams of glucose. To confirm the diagnosis, the patient had to have a glycemic decline below 55 mg/dL during the five-hour test with coexisting symptoms attributed to hypoglycemia. As a result of the performed Oral Glucose Tolerance Test (OGTT), the diagnosis was confirmed in only 30% of cases. However, among all remaining patients from the study group, symptoms attributed to hypoglycemia were observed, although the glucose value was not lower than 55 mg/dL at any time-point. Because the diagnosis of RH is not standardized (lack of standardized biochemical test and lack of glucose cut-off value), the study group comprised participants who experienced symptoms attributed to hypoglycemia in the postprandial period, with or without confirmed reactive hypoglycemia diagnosis, interested in long-term dietary intervention supported by a registered dietitian. Any metabolic disorder; liver, heart, or kidney failure; diagnosed tumors of pancreatic island; oncological diseases; sepsis; unrivalled endocrine disorders; condition after stomach or bowel resection; active stomach ulcer disease; pregnancy or menopause; treatment with drugs that may cause hypoglycemia; and alcohol abuse were listed as the exclusion criteria.

The study was performed in the Central Clinical Hospital of the Ministry of Internal Affairs and Administration (MSWiA) in Warsaw, Poland. The first consultation took place in the Department of Internal Medicine, Endocrinology and Diabetology, and the follow-up consultations were performed in the Metabolic Outpatient Clinic in MSWiA Hospital.

Forty subjects (thirty-three women and seven men) were enrolled to the study and all the participants completed the trial. None of the participants quit or were excluded from the trial due to non-adherence (non-attendance at the follow up appointments or failure to implement recommendations). This is the very first study conducted for this purpose and, given the relatively small number of participants, it should be considered as a preliminary study.

Participants obtained information on the purpose and long-term duration of the study. All subjects signed their written informed consent. The study was approved by the Commission for Ethics and Supervision of Human and Animal Research at the Central Clinical Hospital of the Ministry of Internal Affairs and Administration in Warsaw (protocol code E.D/33/2019 and date of approval 25 January 2019).

### 2.2. Study Protocol

The study was conducted between 2019 and 2021. The dietary interventions included two healthy diet models—LGID and MD. Each subject was introduced to a particular diet via one-on-one education performed by a registered dietitian. Each patient was advised about recommended and unrecommended meal choices based on the food pyramid, charts, and detailed guidelines. Every diet had to be implemented for three months with one follow up consultation in the middle. A wash-out period between the two diets was not performed. There was no specific target for daily intake of calories and nutrients. The patients did not have to perform any calorie counting.

During the study, participants were responsible for their own dietary choices such as food purchases or food preparation. All participants were required to follow the rules and instructions given during dietary consultations. Patients were told to maintain their usual level of physical activity. After six months under dietitian supervision, the patients had no imposed dietary patterns and were able to make individual dietary choices based on their own food preferences.

The schema chart provided in Figure 1 summarizes the study protocol.

### 2.3. Evaluation of Dietary Habits

Each patient took a standardized, original dietary questionnaire based on Dietary Habits and Nutrition Beliefs Questionnaire designed by Behavioral Nutrition Team Committee of Human Nutrition, Polish Academy of Sciences [24]. The questionnaire focused on the daily dietary habits and frequency of consumption of a particular group of products. In terms of frequency, patients could choose the following answers: “a few times a day”, “once a day”, “a few times a week”, “once a week”, “1–3 times a month”, and “never”.

The dietary questionnaire was completed three times. The first evaluation took place after enrolment into the study and considered dietary habits before the research commenced. The second was completed after six months and evaluated eating habits following a regular dietary supervision. The final one was completed after twelve months from the last follow-up appointment and looked at patients’ current dietary practices.

### 2.4. Dietary Guidelines Adherence Index

The Dietary Guidelines Adherence Index (DGAI) was used to assess patients’ adherence to the eating pattern recommendations and to evaluate the diet’s quality index in terms of healthy and unhealthy food choices based on the frequency of food consumption. This tool has provided an assessment of dietary habits over a three-point period—before the dietary intervention, at the end of the dietitian supervision phase, and after twelvemonths with no imposed dietary patterns. The original DGAI has been enhanced with additional food products defined as healthy or unhealthy [25]. The first group included healthy products that should be consumed habitually as they exerted a positive effect on health, such as: vegetables, fruits, whole grains (bread, groats, rice, pasta, oatmeal)**,** dairy products (no added sugar), eggs, legumes, fish, lean meat, unsalted nuts, seeds, olive oils, and avocado. The group of unhealthy foods included excessive amounts of red meat, processed meat, sweetened dairy products, processed cheese, butter, margarine, refined grains (bread, groats, rice, pasta, sweetened breakfast cereals)**,** fast food, and sweets. The evaluation was based on the New Pyramid for a Sustainable Mediterranean Diet [26].

The rating scale remained unchanged from the initial version of the questionnaire—the higher index value that was obtained, the greater degree of adherence to dietary recommendations was observed.

### 2.5. Previous Diets and Current Preferable Dietary Pattern

Before the dietary intervention began, each participant had been surveyed about the previous diets that were chosen to reduce postprandial symptoms attributed to hypoglycemia. After completing a series of dietary consultations, the patient was asked to determine which dietary model (LGID vs. MD) was more convenient in terms of daily meal preparation. The evaluation investigated whether the previously adopted diets had any influence on the current preferable dietary pattern.

### 2.6. Evaluation of Symptoms Attributed to Hypoglycemia

Each patient underwent a standardized, original questionnaire in terms of symptoms attributed to hypoglycemia. The subjects have rated the severity of symptoms such as: sweating, hunger, tachycardia, tremor, anxiety, fatigue, blurred vision, impaired concentration, slurred speech, and fainting, which may occur in the postprandial period. Each hypoglycemic symptom was rated using a numerical scale, for which 1 represented the absence of symptoms and 5 indicated very high severity of hypoglycemic-like symptoms. This assessment was a subjective perception of each patient. This questionnaire was administered four times—in the pre-study phase, at the completion of each diet phase, and after twelve months with no dietary supervision.

## 3. Statistics

Quantitative and categorical (nominal and ordinal) variables were described with descriptive statistics methods. For quantitative variables, the following measures were determined: central tendency (mean, M) and dispersion (standard deviation, SD). For categorical variables the following measures were determined: number (N) and frequency (%).

Cross tables with odds ratio (OR) and a two-tailed Fisher’s exact test or Cochran’s Q test were used to compare the selected nominal variables. The choice of the test was determined by the design: independent or dependent samples, respectively. Additionally, Cochran’s Q test results were calculated along with post hoc test with the Bonferroni adjustment. The Friedman test was used to compare the ordinal variables in the dependent samples. Additionally, Friedman test results were calculated along with post hoc Dunn’s test.

An analysis of variance (ANOVA) for repeated measures was used to compare the mean values of the healthy and unhealthy index at three time points. The ANOVA was supplemented with the calculation of the effect size (eta-squared with 90% confidence interval (CI)) and Fisher’s least significant difference (LSD) post hoc test. Skewed variables were transformed by the Box–Cox method to obtain a symmetrical distribution (skew in the range from −0.5 to +0.5).

All calculations were performed with STATISTICATM 13.3 software (TIBCO Software, Palo Alto, CA, USA). For all analyses, a *p*-level of <0.05 was considered statistically significant.

## 4. Results

Forty adults (mean age 37.0 ± 9.9 years) with correct body mass index (mean BMI 23.7 ± 3.0 kg/m^2^) who reported symptoms attributed to hypoglycemia were enrolled in the study. The official clinical diagnosis (glucose decline below 55 mg/dL and coexisting symptoms during five-hour OGTT) of RH was not a requirement for the inclusion into the study. However, twelve patients (nine women, three men) had confirmed diagnosis, whereas the other twenty-eight participants (twenty-four women, four men) did not meet the diagnostic criteria (presence of hypoglycemic-like symptoms without glucose decline below 55 mg/dL during five-hour OGTT).

All the subjects completed the entire study protocol.

### 4.1. Previous Dietary Experience

The Dukan diet (40%), vegetarian diet (35%), and gluten-free diet (32%) were the most frequently selected dietary patterns to reduce postprandial hypoglycemic-like symptoms. The lactose-free diet (12%), monotrophic diet (8%), and Atkins diet (3%) were chosen the least. (Figure 2). None of the implemented diets had any effect on symptoms reduction.

### 4.2. Number of Main Meals per Day

The analysis of the number of consumed meals per day revealed a significant difference (χ^2^_(*df*=2)_ = 17.577, *p* < 0.001) in the three-point period: baseline, after LGID and MD, and twelve months from the dietary supervision (Figure 3). The number of patients who preferred only three meals per day has decreased in a three-fold manner—both immediately after the end of the dietary supervision (10%) and twelve months after the dietary appointments (7.5%) in comparison to the pre-intervention period (30%). The number of patients who chose five meals has doubled after a six-month period of dietary supervision (25%) compared with the baseline (12.5%). Consumption of four meals was reported most frequently—in the baseline (50%), immediately after the end of the intervention (62.5%), and with the greatest increase registered after a twelve-month period (75%). Post hoc analysis showed a statistically significant difference in the number of consumed meals per day between the pre-intervention period and immediately after the end of the dietary supervision (z = −0.512, *p* = 0.022). No significant change was observed between the second (after dietary supervision) and the third (twelve months after dietary supervision) time points of analysis (z = 0.125, *p* = 0.576).

### 4.3. Time Gap between Meals

The analysis of the time gap between consumed meals revealed a significant difference (χ^2^_(*df*=2)_ = 37.172, *p* < 0.001) in the three-point period: baseline, after LGID and MD, and twelve months from the dietary supervision (Figure 4). Before the intervention, the majority (57.5%) of patients confirmed having a break between their main meals that was longer than six hours. During dietary supervision, the majority of the patients (60%) consumed meals three to four hours apart. After twelve months this habit has slightly changed—most of the patients (55%) extended the time gap between meals to a five to six hours, and the rest (45%) remained on a three to four hours break pattern. Post hoc analysis showed a statistically significant difference in the time gap duration between meals in pre-intervention period and twelve months from the dietary supervision (z = 0.788, *p* < 0.001). Moreover, significant change was observed between the second (after dietary supervision) and the third (twelve months after dietary supervision) time points of analysis (z = 1.200, *p* < 0.001).

### 4.4. Meal Frequency

Significant differences (Q*_df_*_=2_ = 31.182, *p* < 0.001) occurred in the meal frequency in the three-point period: baseline, after LGID and MD, and twelve months from the dietary supervision. Before the intervention, only 10% of respondents had declared regular consumption. Over the course of the study, nearly two-thirds (62.5%) reported a frequent meals intake, although less than half (45%) continued this habit after the twelve-month period (Figure 5).

Post hoc test with the Bonferroni adjustment analysis showed a statistically significant difference in the frequency of consumed meals between pre-intervention period and immediately after the end of the dietary supervision (z = −0.525, *p* < 0.001) and twelve months after the dietary intervention (z = −0.350, *p* = 0.010). The significant change was observed between the second (after dietary supervision) and the third (twelve months after dietary supervision) time points of analysis (z = 0.175, *p* = 0.203).

### 4.5. Snacking between Meals

Significant differences occurred in terms of snacking between the main meals (χ^2^_(*df*=2)_ = 53.660, *p* < 0.001) in the three-point period: baseline, after LGID and MD, and twelve months from the dietary supervision. More than half of the participants (57.5%) used to snack once a day before the intervention, which significantly decreased after six months of dietary supervision (5%) and twelve months from the dietary supervision (5%). Over the course of the study, patients mostly reported snacking once a week (40%), however after twelve months this habit enlarged to several times a week (40%) (Table 1). Post hoc analysis showed a statistically significant difference in the frequency of snacking between the pre-intervention period and immediately after the end of the dietary supervision (z = 1.213, *p* < 0.001) and twelve months after the dietary intervention (z = 1.000, *p* < 0.001). No significant change was observed between the second (after dietary supervision) and the third (twelve months after dietary supervision) time points of analysis (z = −0.212, *p* = 0.342).

### 4.6. Type of Selected Snacks

Before the intervention, none of the subjects consumed vegetables as a type of snack. This habit significantly improved after the dietary intervention and continued after 12 months (31% and 33%, respectively). Vegetables became the most common type of snack, followed by nuts, seeds, and fruits. Just after the dietary supervision and after twelve months, participants excluded sweetened dairy products, sweets, and salty snacks as a type of snack (Table 2).

### 4.7. Hydration

In all three periods (baseline, after LGID and MD, and twelve months from the dietary supervision), patients selected water and tea as the most common type of fluids. There was a significant reduction in juice consumption (χ^2^_(*df*=2)_ = 8.714, *p* < 0.001) after dietary intervention. Post hoc test with the Bonferroni adjustment showed no significant change between the second (after dietary supervision) and third (twelve months after dietary supervision) time points (z = −0.187, *p* = 0.402).

Significant differences occurred in terms of added sugar in warm beverages (Q*_df_*_=2_ = 22.000, *p* < 0.001). After six months of dietary supervision and after twelve-months follow-up, patients entirely abandoned added sugar, which was used by 30% of the subjects in the baseline (Figure 6). Post hoc test with the Bonferroni adjustment showed a statistically significant difference in the amount of added sugar between pre-intervention period and immediately after the end of the dietary supervision (z = −0.275, *p* < 0.001) and after twelve months after the dietary intervention (z = −0.275, *p* < 0.001). No significant change was observed between the second (after dietary supervision) and the third (twelve months from the dietary supervision) time points of analysis (z = 0.000, *p* = 1.000).

### 4.8. Dietary Guidelines Adherence Index

The analysis of the adherence to healthy and unhealthy patterns evaluated via the Dietary Guidelines Adherence Index showed significant differences (F_(2,78)_ = 27.30, *p* < 0.001, η^2^ = 0.41, 90%CI [0.26; 0.51]) in the three-point period. (Figure 7). The healthy eating pattern improved over the period of the dietary consultations and remained at the same level after twelve months. Fisher’s least significant difference (LSD) post hoc test showed a statistically significant difference in the healthy eating pattern between the pre-intervention period and immediately after the end of the dietary supervision (*p* < 0.001) and twelve months after the dietary intervention (*p* < 0.001). No significant change was observed between the second (after dietary supervision) and the third (twelve months after dietary supervision) time points of analysis (*p* = 0.938).

The unhealthy diet index decreased significantly (F_(2,78)_ = 433.39, *p* < 0.001, η^2^ = 0.91, 90%CI [0.89; 0.93]) during the period of dietary supervision compared to the baseline period (Figure 8). After twelve months the index slightly increased, but it was not a statistically significant difference (*p* = 0.060) in overall evaluation. The non-substantial rise was the result of a greater consumption of highly processed foods such as white bread, breakfast cereals, sweets, and junk food compared to the time of dietary supervision. Fisher’s least significant difference post hoc test showed a statistically significant difference in the unhealthy eating pattern between pre-intervention period and immediately after the end of the dietary supervision (*p* < 0.001) and after twelve months after the dietary intervention (*p* < 0.001).

### 4.9. Hypoglycemic-like Symptoms in Postprandial Period

There was a statistically significant reduction in the severity in eight out of ten analyzed hypoglycemic-like symptoms after dietary interventions (Table 3). The strongest effect was observed for the following symptoms: hunger (η^2^ = 0.66), impaired concentration (η^2^ = 0.61), hand tremor (η^2^ = 0.55), and fatigue (η2 = 0.51). Blurred vision and slurred speech were the least reported symptoms and no significant disparities were observed in all four evaluations (at baseline, after LGID, after MD, and twelve months from the dietary supervision). The occurrence of the hypoglycemic-like symptoms was comparable between the first diet (LGID) and the second nutritional strategy (MD). However, while following the MD, patients reported greater reduction in the severity of the following symptoms: hunger (*p* < 0.001), tremor (*p* < 0.001), anxiety (*p* < 0.001), and impaired concentration (*p* < 0.001).

### 4.10. Preferable Dietary Pattern

At the end of the dietary supervision, a significant percentage of patients (80%) selected the Mediterranean diet as a preferable dietary pattern (Figure 9). Previous dietary regimens had no effect on the currently selected dietary pattern (Table 4).

## 5. Discussion

It should be emphasized that our research is the first, long term study that analyzed the severity of symptoms attributed to hypoglycemia in line with dietary intervention. The study, conducted in the real-world setting, has shown that adequate dietary interventions might have a profound impact on the reduction of hypoglycemic-like symptoms in the postprandial period in non-diabetic patients.

The dietary survey conducted at the pre-study stage allowed us to assess the habitual nutritional habits of subjects with symptoms attributed to hypoglycemia. Most patients used to have four meals per day without any regular pattern in terms of meal timing. The time gap between meals was greater than six hours; nonetheless, the majority of patients reported additional snacking once a day. Half of the respondents reported frequent consumption (several times a week) of red meat and meat products, whereas fish appeared in the diet only occasionally (one to three times a month). On a weekly basis, sweetened dairy products and high-fat dairy products were preferred over natural, skim, and semi-skimmed dairy products. Refined grains appeared to be chosen notably more often than whole-grain alternatives. Throughout the day, fruits were consumed more frequently than vegetables. In addition, 50% of patients consumed fast food once a week. Sweets were eaten once a day by 60% of the subjects, and 25% of participants reported eating candies several times a day. These observations presented a view of the typical Western diet, which is characterized by a high intake of highly processed foods, rich in sugars and saturated fats.

Moreover, the majority of participants reported undertaking various dietary modifications in order to reduce hypoglycemic-like symptoms. This may suggest a potential public awareness of the notable nutritional impact on comorbidities and overall well-being [27]. However, insufficient dietary knowledge and lack of support from an experienced dietitian can lead to the implementation of unsuitable dietary modifications [28]. The patients testified that selected dietary adjustments were self-initiated without the professional assistance of a dietitian. The dietary choices were diverse and included elimination diets (gluten-free, lactose-free), plant-based diets (vegetarian, vegan) and diets with modified macronutrient intake (Ketogenic, Atkins, Paleo, and Dukan). The controversial, low-energy diets such as the Copenhagen diet or the monotrophic diet (the monodiet or single-food diet), which could lead to nutrient deficiencies, also were reported by the patients. The most prevailing diet appeared to be the Dukan diet, characterized as a high-protein, low carbohydrate diet based primarily on meat, poultry, fish and seafood, eggs, and dairy. In all cases, chosen modifications did not improve well-being and failed to reduce the severity of hypoglycemic-like symptoms. There is no scientific research to support the non-beneficial outcome of given diets in terms of reactive hypoglycemia. Nevertheless, assuming that those diets may have been poorly balanced with insufficient nutrient quantity, the lack of effectiveness should not be questioned.

The current study has focused on two types of diets—LGID and MD, commonly established as healthy eating patterns [29,30]. Interestingly, before the study, none of the participants had implemented these nutritional habits as a part of a diet therapy. It should be highlighted that LGID and MD also have not been appraised in any scientific studies in the context of hypoglycemic-like symptoms in non-diabetic individuals. Therefore, no other methodical evidence can be invoked in the direct relation to the diet composition in line with suspected RH. In contrast to the previous nutritional alterations that were self-implemented, the dietary adjustments proposed by the dietitian resulted in positive changes in overall well-being. The effectiveness of the nutritional modifications should be considered in terms of the individual dietary approach, the adjustments to the daily nutrition, patient’s adherence, and psychological factors.

The first major part was the one-on-one dietary education that each patient underwent. In this manner, it was possible to share all the essential information about the diet, focusing on the list of recommended and non-recommended foods, frequency of consumed meals, and proper hydration. Each patient had the opportunity to ask questions relating to the diet modifications, which ensured a better understanding of the recommendations. This is an indispensable principle to plan and prepare the meals correctly, respecting the guidelines and dietary preferences. Tailoring the healthy diet to match dietary preferences and nutritional requirements is one of the key approaches to enhance adherence to the dietary changes [31]. When deciding on the sequence of diets (LGID and MD), it was considered that patients tend to be more committed in the first stage of alterations, which may diminish over time [32]. The low glycemic index diet was proposed as the first dietary pattern, considering that the diet may be more demanding due to the need to analyze the glycemic index value of chosen products. The second dietary protocol, based on the recommendations of the Mediterranean diet, intended to be a continuation of the well-established dietary habits but with a less restricted approach. The follow-up appointments were designed to verify the implemented changes and to sustain the patient’s self-engagement. It appears to be the most reliable approach to support consistency in the everyday diet [33]. After a period of six months, two educational meetings, and two follow-up appointments, significant improvements in the dietary habits in line with a decline in the severity of hypoglycemic-like symptoms were observed. Moreover, the reported decrease in symptoms was comparable after LGID and MD. Based on the study results, the outcome was promising, particularly in the perspective of previous failures in the effectiveness of dietary modifications undertaken by the patients. It is reasonable to assume that the proper understanding of the healthy dietary guidelines and the perceived improvement in a well-being may have prompted the willingness to continue with the initiated dietary changes. To verify this assumption, the follow up was performed after twelve months. A one-year follow-up seemed to be sufficient duration that may demonstrate a potential shift in the dietary patterns. Mixed outcomes were expected—on one side, adherence to the recommendations that had improved overall well-being, but on the other, deterioration of healthy habits due to the lack of dietary check-ups. The final report obtained after twelve months confirmed the participant’s perseverance to follow the beneficial eating habits in line with sustained tendency in the decline in symptom severity in the postprandial period.

The healthy diet index remained unmodified after one year in comparison to the period of dietary supervision. Patients were more likely to choose whole grains, vegetables, foods that provide a good source of protein (lean meat, natural dairy products, legumes), and healthy fats (nuts, seeds, fish, vegetable oils) compared to the baseline. The long-term dietary interventions resulted in a positive outcome in terms of the correct number of meals consumed per day both during and after the dietary supervision. This might be consistent with the statement that shows an association between improvement in the diet quality and larger number of smaller meals [34]. The regularity of meals was found to be better during the intervention than in the post-intervention period; however, no increase in the frequency of snacking was observed. The significant reduction in snacking in-between meals, in comparison to the pre-intervention period, may indicate a greater awareness of choosing well-balanced meals that provide a longer satiety duration [35]. Moreover, improved dietary knowledge helped to promote a shift in a snack type from highly processed sweets to low-processed foods, such as vegetables. Considering the presentation of symptoms (typical for hypoglycemia) in the postprandial period, the idea of a well-structured meal plan, which would enforce regularity of meals consumption, seems to be favorable. This could diminish a sudden onset of hunger, irritation, or impaired concentration, which are identifiable symptoms consistent with hypoglycemia. In addition, proper meal timing would lower the temptation of snacking, which could be favorable in the reducing a sudden glucose fluctuation.

Interestingly, the unhealthy diet index did not change significantly after one year; patients were more likely to consume highly processed foods. Those choices did not increase the severity of hypoglycemic-like symptoms, although the adverse effects may be noticed at a posterior period. It should be noted that sustainable dietary changes require a strong personal commitment, which may be challenging in the long-term outlook. For this reason, follow-up appointments and support from a dietitian may be essential in preserving prolonged improvements [36]. The frequency of consultations should be individually arranged, based on the patient’s needs.

In light of selecting the best dietary pattern for patients with hypoglycemic-like symptoms, the proposed models seem to have beneficial effects. Adherence to LGID and MD had significant impact in terms of symptoms reduction, which supports their effectiveness. In assessing the efficiency of those two diets, it is relevant to underline their common factor—low-processed and high-fiber foods. The main difference between those dietary patterns is the ability to choose healthy products with higher glycemic index, which is acceptable in the Mediterranean diet. The favorable effects of MD may undermine the exclusion of medium- and high-glycemic index foods, particularly, if a patient with hypoglycemic-like symptoms does not have insufficient secretion of insulin or impaired insulin action. It should be highlighted that glycemic index may affect the postprandial glycemia; however, non-diabetic patients do not indicate a negative association with changes in β-cell function or oxidative stress markers [37]. Based on the reported symptom decline that was achieved via both dietary models, it seems that there is no evidence to avoid healthy low-processed foods but with higher glycemic index. Therefore, the improvement in well-being should not be solely explained by the blood glucose stabilizing effects, as it appears that glycemic fluctuations may not be the primary causative factor.

Most dietary studies are appraised in the view of individuals with comorbidities. Thus, the impact of the implemented diet can be evaluated in terms of physiological and biochemical alterations. In the case of patients without metabolic disturbance but still manifesting symptoms attributed to hypoglycemia, it is difficult to clearly explain the changes observed after the dietary modifications. However, there is a remarkable amount of research demonstrating the influence of a dietary pattern on improvement of overall health and well-being [38,39,40]. It is reasonable to speculate whether a better state of mind may have reduced the perception of symptoms attributed to hypoglycemia. The individual, long-term supervision has allowed for focusing on the patient’s concerns and collectively discussing dietary changes in line with symptom frequency. This may have had an impact on better understanding of the patient’s health status, which influences the improvement of psychological well-being. Therefore, non-nutritional factors, such as emotional and psychological states, which may affect symptoms severity, should also be taken into consideration during the treatment. Hence, along with dietary assistance, the patient should be supported with psychological care.

Considering the obtained results that emphasize the positive impact of a healthy eating pattern on overall well-being, it is also important to highlight the constantly debated topic of the gut microflora and its influence on overall health [41]. There is an interrelated link between nutrition and the microbiota because dietary habits influence colonization and life-long microbiome rearrangement. A pilot study involving patients with RH evaluated the composition of the gut microbiota and the potential role of a diet rich in fiber and fermented products [42]. The short-term observation confirmed the positive impact of diet composition on the increase in short-chain fatty acids (SCFAs), which are end products of the microbiota fermentation of complex carbohydrates in the gut. CFAs are involved in many metabolic pathways, including glucose metabolism. Moreover, there is a growing number of studies investigating the correlation between the composition of the gut flora and mental health. Based on these observations, it may be speculated whether the improvement in the dietary habits also has a beneficial impact on the microflora, which, in turn, may have promoted better state of mind among the patients [43]. However, this was not the intended purpose of our study, and none of those parameters were assessed during the research. A further investigation in this regard would be recommended in order to explore more specific dietary findings in line with symptoms attributed to hypoglycemia.

Several limitations should be considered in the interpretation of the study. The first weakness includes evaluation of hypoglycemic-like symptoms. Perceived symptoms may have subjective interpretation and be affected by many additional factors. The rating scale allows the patient to determine the severity of the symptoms, but the risk of over- or under-scoring by the patient should be considered. The second limitation is the reliability of the responses regarding the dietary habits. The data were analyzed based on self-reported answers given by the participants. It raises the question of recall bias and potential modification of the actual dietary choices only to match the expectations. However, based on the study results, improvements in dietary habits were consistent in the long-term perspective and in line with symptom decline reported by the patients. It may suggest a reliable dietary adherence and accuracy of given answers. A third limitation is the lack of a control group. One of the main purposes of this research was to evaluate the occurrence of postprandial hypoglycemic-like symptoms in line with dietary modifications. In healthy subjects or those without suspected reactive hypoglycemia, the incidents indicating hypoglycemia are not commonly observed. Therefore, it would not be possible to evaluate the impact of dietary modifications on the onset of hypoglycemic-like symptoms. A final limitation is the fact that not all disease entities could be diagnosed during the performed screening. Despite thorough medical examination and exclusion of many pathological conditions, there is a risk of under diagnosis. Rare conditions such as pheochromocytoma can also lead to hypertension (chronic or paroxysmal) and present similar symptoms to those associated with RH. However, the participants had no history of hypertension and multiple measurements taken in the hospital setting confirmed blood pressure was correct. Additionally, reduction of symptoms after the implementation of dietary modifications would further suggest that pheochromocytoma diagnosis is highly unlikely.

Given the fact the study itself may have been the initial investigation of this issue, the reproducibility of the study cannot be assessed unambiguously. Moreover, this is the first study conducted in this field and, due to the relatively small number of participants, it should be treated as a preliminary study. It should be noted that the study is a long-term investigation performed in a real-world setting, therefore the reliability of the obtained results tends to be higher and have the potential to capture outcomes that are more relevant to the patients. For this reason, it is highly recommended to perform equivalent research, with the applied methodology on a larger group of patients.

In conclusion, based on the results, the study demonstrates that among patients with symptoms attributed to hypoglycemia, there was significant improvement in the reduction of symptoms consistent with hypoglycemia after dietary intervention based on a healthy eating pattern. The evaluated effects of LIGD and MD were similar; however, participants rated the Mediterranean diet as a pattern that was easier to follow in the daily life and several symptoms became less severe. Hence, it is reasonable to believe that more than just one dietary factor could be responsible for the improvement in patients’ well-being. The study approach was to achieve substantial changes in the overall eating patterns that were both realistic and had beneficial outcomes. This purpose has been accomplished in light of the long-term dietary adjustments and the subsequent decrease in symptoms attributed to hypoglycemia. Therefore, greater attention should be devoted to the one-on-one dietary education based on a healthy eating pattern rather than the macronutrient composition of the diet per se. Furthermore, it should be recommended to provide long-term support for the patient in the process of dietary changes—both nutritionally and psychologically. Due to the favorable outcome of the study, further research is recommended to investigate a relationship between dietary patterns and reduction of symptoms attributed to hypoglycemia.

## 6. Conclusions

Healthy modification in the dietary habits may improve the well-being and reduce the severity of postprandial hypoglycemic-like symptoms. Therefore, dietary intervention focused on appropriate nutritional management combined with follow-up consultations may be a beneficial step towards comprehensive treatment of non-diabetic patients who present with hypoglycemic-like symptoms in the postprandial period.

## Figures and Tables

**Figure 1 nutrients-14-00497-f001:**
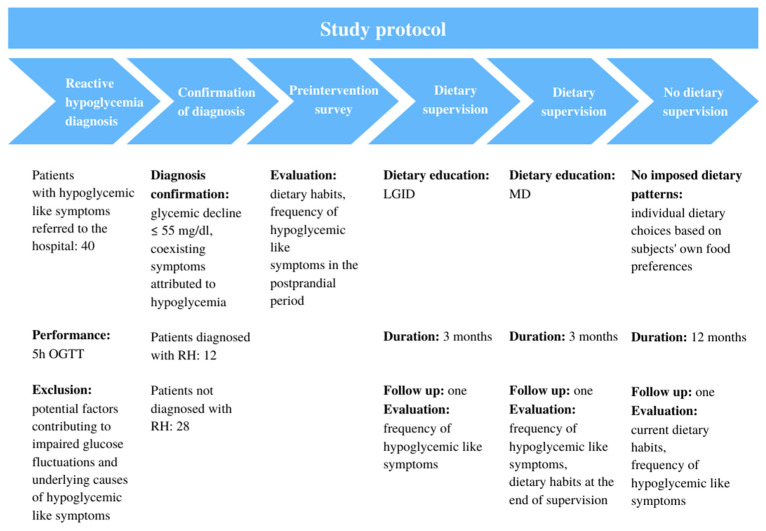
Study protocol. OGTT—oral glucose tolerance test; RH—reactive hypoglycemia; LGID—low glycemic index diet; MD—Mediterranean diet.

**Figure 2 nutrients-14-00497-f002:**
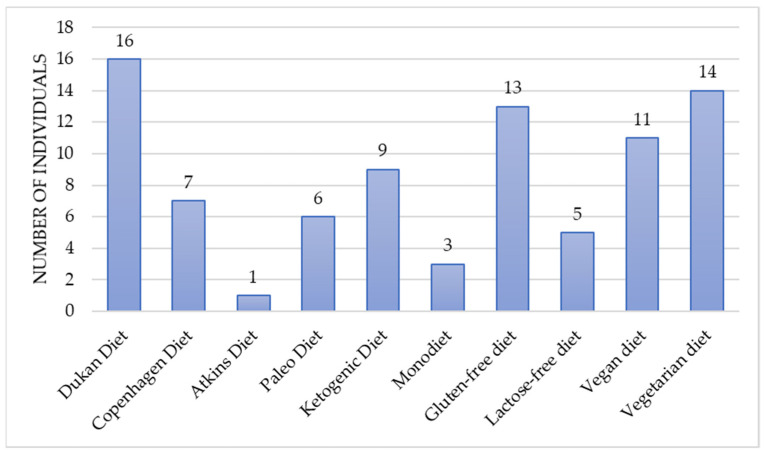
Previously implemented diets to reduce symptoms attributed to hypoglycemia.

**Figure 3 nutrients-14-00497-f003:**
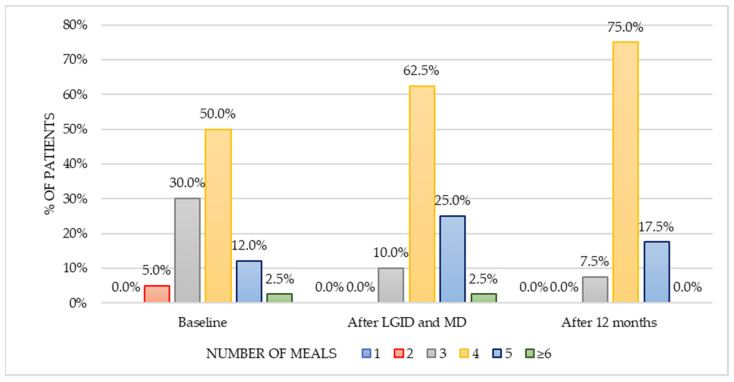
Number of main meals consumed by the patients per day in three-point period.

**Figure 4 nutrients-14-00497-f004:**
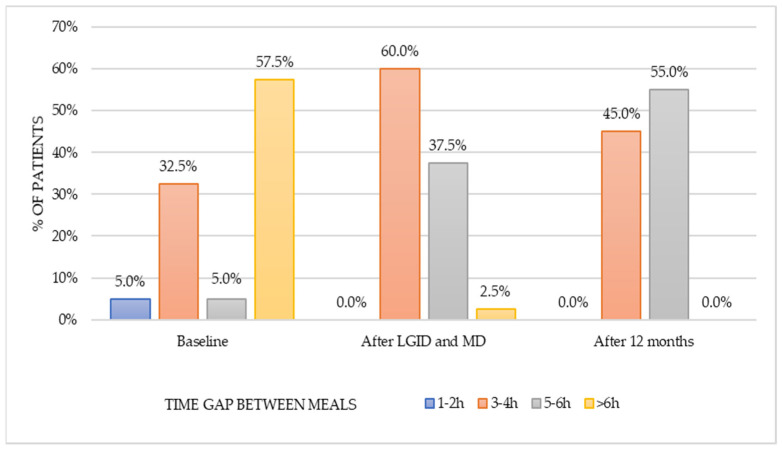
Time gap between meals in the three-point period.

**Figure 5 nutrients-14-00497-f005:**
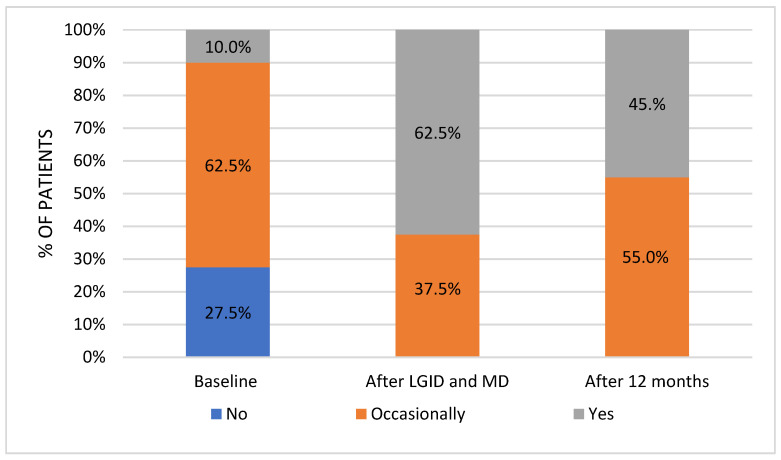
Meal frequency in the three-point period.

**Figure 6 nutrients-14-00497-f006:**
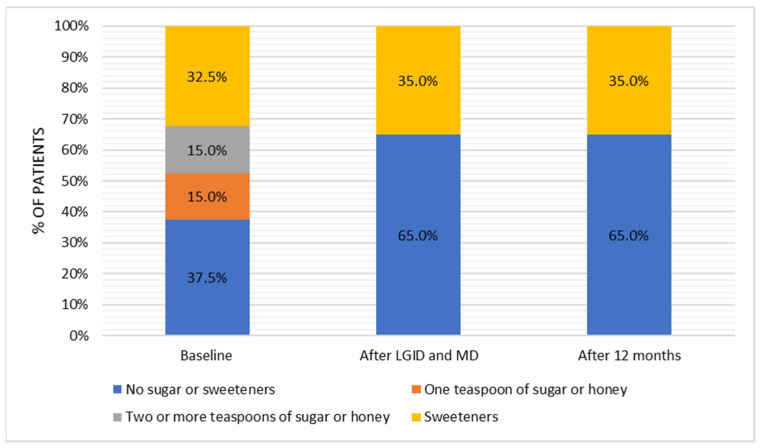
Adding sugar to consumed beverages in the three-point period.

**Figure 7 nutrients-14-00497-f007:**
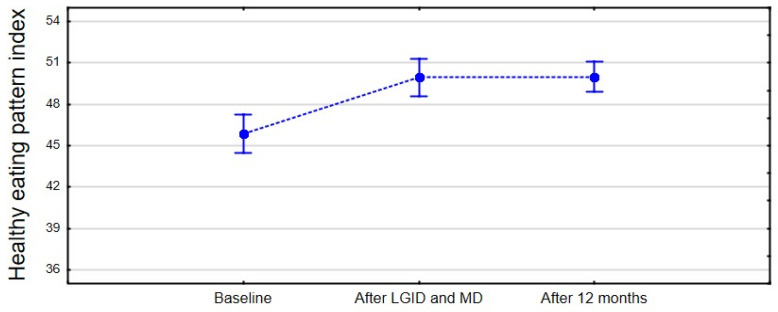
The mean of the healthy eating pattern index in the three-point period.

**Figure 8 nutrients-14-00497-f008:**
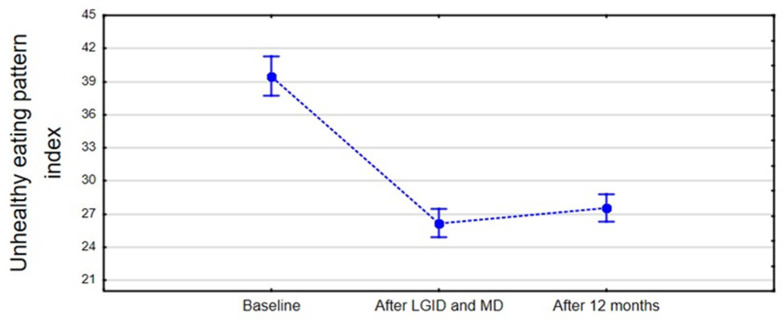
The mean of the unhealthy eating pattern index in the three-point period.

**Figure 9 nutrients-14-00497-f009:**
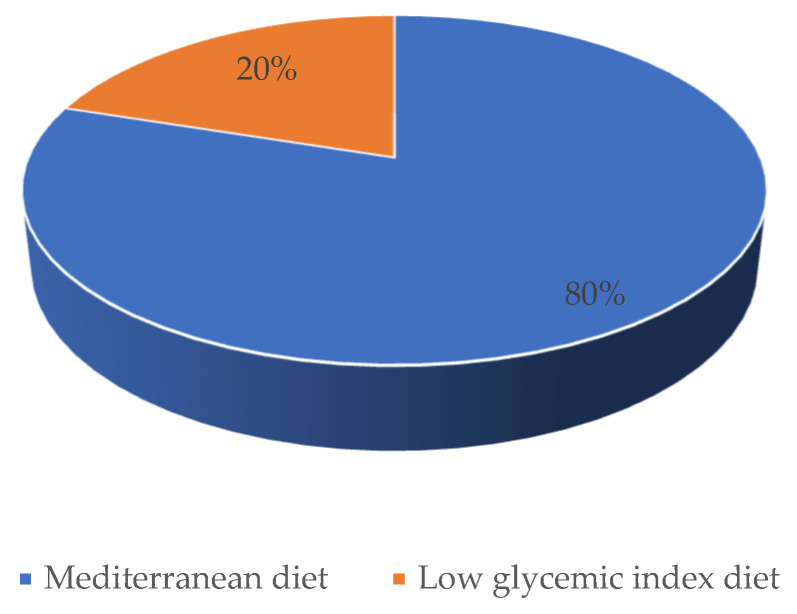
Preferable dietary pattern after six months dietary supervision.

**Table 1 nutrients-14-00497-t001:** Frequency of snaking in the three-point period.

Frequency	Baseline		After LGID and MD		After 12 Months		χ^2^_(*df*=2)_	*p*-Value *
	N	%	N	%	N	%		
Never	8	20.0	8	20.0	7	17.5	53.660	<0.001
1–3 times a month	0	0.0	3	7.5	2	5.0		
Once a week	0	0.0	16	40.0	13	32.5		
Few times a week	3	7.5	11	27.5	16	40.0		
Once a day	23	57.5	2	5.0	2	5.0		
Few times a day	6	15.0	0	0.0	0	0.0		

* Friedman test.

**Table 2 nutrients-14-00497-t002:** Type of consumed snacks in the three-point period.

Types of Snacks	Baseline		After LGID and MD		After 12 Months	
	N	%	N	%	N	%
Fruits	15	46.9	7	21.9	7	21.2
Vegetables	-	-	10	31.3	11	33.3
Unsweetened dairy products	3	9.4	5	15.6	5	15.2
Sweetened dairy products	3	9.4	-	-	-	-
Sweet snacks	6	18.8	-	-	-	-
Salty snacks	1	3.1	-	-	-	-

**Table 3 nutrients-14-00497-t003:** Hypoglycemic symptoms in postprandial period.

Symptom	Baseline		After LGID		After MD		After 12 Months		F*_df_*_=3_	*p*-Value *	η^2^ (90%CI)
	M	SD	M	SD	M	SD	M	SD			
Sweating	1.63	0.95	1.45	0.81	1.43	0.64	1.43	0.64	3.201	0.026	0.08 (0.01; 0.14)
Hunger	3.88	0.88	2.35	0.74	1.78	0.58	2.03	0.66	76.007	<0.001	0.66 (0.57; 0.71)
Tachycardia	1.18	0.55	1.08	0.35	1.05	0.22	1.05	0.22	3.266	0.024	0.08 (0.01; 0.15)
Tremor	3.45	1.08	2.15	0.83	1.75	0.63	1.70	0.56	48.101	<0.001	0.55 (0.44; 0.62)
Anxiety	3.38	0.90	2.78	0.83	2.08	0.53	2.10	0.55	40.028	<0.001	0.51 (0.39; 0.58)
Fatigue	3.23	1.05	2.45	0.78	2.23	0.48	2.28	0.51	26.633	<0.001	0.41 (0.28; 0.49)
Blurred Vision	1.05	0.32	1.05	0.32	1.03	0.16	1.03	0.16	1.000	0.396	-
Impaired Concentration	4.20	0.85	3.00	0.78	2.38	0.63	2.43	0.55	60.153	<0.001	0.61 (0.51; 0.67)
Slurred Speech	1.08	0.47	1.08	0.47	1.05	0.32	1.00	0.00	1.000	0.373	-
Fainting	1.60	1.03	1.05	0.22	1.15	0.58	1.08	0.27	9.935	<0.001	0.20 (0.09; 0.29)

M—mean, SD—standard deviation, *df*—degree of freedom, η^2^—eta-squared. * Repeated measures ANOVA.

**Table 4 nutrients-14-00497-t004:** Influence of previous diets on choice of current preferred dietary pattern.

Previous Diets	MD (N = 32)		LIGD (N = 8)		*p*-Value *	OR (95%CI)
	N	%	N	%		
Dukan Diet	15	46.88	1	12.50	0.114	0.16 (0.02; 1.47)
Copenhagen Diet	5	15.63	2	25.00	0.611	1.80 (0.28; 11.60)
Atkins Diet	1	3.13	0	0.00	1.000	1.40 (0.05; 37.89)
Paleo Diet	6	18.75	0	0.00	0.318	0.24 (0.01; 4.71)
Ketogenic Diet	8	25.00	1	12.50	0.655	0.43 (0.05; 4.04)
Monodiet	3	9.68	0	0.00	1.000	0.48 (0.02; 10.22)
Gluten-free diet	11	35.48	2	25.00	0.694	0.61 (0.10; 3.53)
Lactose-free diet	5	15.63	0	0.00	0.563	0.29 (0.01; 5.88)
Vegan diet	10	31.25	1	12.50	0.405	0.31 (0.03; 2.91)
Vegetarian diet	13	40.63	1	12.50	0.222	0.21 (0.02; 1.90)

OR—odds ratio, CI—confidence interval. * Two-tailed Fisher’s tests.

## Data Availability

The data presented in this study are openly available.

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
