# Peer review of "Assessing Long-Term Impact of Dietary Interventions on Occurrence of Symptoms Consistent with Hypoglycemia in Patients without Diabetes: A One-Year Follow-Up Study"

_nutrients, 2022, doi:10.3390/nu14030497_

Round 1

Reviewer 1 Report

I think that this manuscript is written well. However this manuscript will need to revised just a little bit.

With regard to Introduction

I didn't understand the hypothesis of this study. Please specify your hypothesis. 

With regard to study design 

I think that the reproducibility of this study needs to be clearly stated. If it cannot be stated, it should be stated as the limit of research. 

With regard to  discussions 

Please consider based on this result. In addition, could you consider minimize citations in the manuscript. 

Are there any other study limitations? 

Reviewer 2 Report

  1. The quality of the English language is mediocre and calls for the overall revision of the text by a native English speaker. At present, phrases and expressions throughout the manuscript are unclear, contain clumsy English constructs, typing or syntactic errors (e.g., grammar disagreements, wrong tense and mode of verbs, incorrect use of articles and prepositions, inversions between adjectives and adverbs, unnatural phrase topic, etc.). This often makes the text difficult to understand, so a thorough rewording is needed.
  2. The manuscript title, Abstract and the Introduction section focus on the issue of glucose fluctuations-related symptoms, while details at the beginning of the Results section suggest an idiopathic postprandial syndrome in most participants instead. The selection of patients based on the simple clinical frame (the postprandial pattern of symptoms) is greatly questionable. We must not forget that hypoglycaemic-like adrenergic symptoms do not necessarily have a cause-effect relationship to blood glucose variations. For instance, dietary protein intake has been speculated to induce some cases of idiopathic postprandial syndrome. Moreover, adrenergic discharges from a pheochromocytoma, sometimes resembling the clinical manifestations of hypoglycaemia, are also possible in the postprandial period due to increased intraabdominal pressure. The authors should decide first on the real, unifying basis that underlies the medical profile of their study subjects and then suitably adapt the entire manuscript (title and Discussions included) to this perspective.
  3. In the Material and Methods section, inclusion criteria are scarce and unclear. Was the diagnosis of reactive hypoglycaemia (RH) confirmed? If so, which were the medical investigations that allowed its confirmation?
  4. The questionnaire used to evaluate the so-called hypoglycaemic symptoms should be described in greater detail.

Reviewer 3 Report

Abstract 

  • Consider to shorten the Methods description and focus on the Results. 
  • Provide effect sizes of main findings

Introduction

  • Well described the background, despite the language issues. 

Methods

  • It's unclear how the researchers obtained the details of participants with hypoglycaemic incidence in this study, especially non-diabetic individuals. Please elaborate the recruitment strategy. 
  • Not having a control group is a major flaw that must be addressed or explained. 
  • Consider summarizing the methods in a flow chart for ease of understanding
  • Long term follow-up to gauge the sustainability of both diets is a commendable effort. 
  • Statistical analysis - RM ANOVA was performed. However, the choice should have been based on normality analysis. Was this performed? If yes, please include the statement. Based on the small sample size, I would suspect the distribution to be skewed. 

Results

  • It is interesting to note the popularity of Dukan diet. 
  • Table 1,2 and Figure 1-4 can probably merged into one meaningful table. 
  • Similarly Fig 5 & 6 should appear together for readers to make an easier comparisons

Discussion / conclusion

  • No further comments

References

  • Appropriate and reasonably current

Round 2

Reviewer 3 Report

Abstract 

I would disagree with the response. Please reconsider to shorten the methods. The explanation given may be valid for the actual manuscript text. The abstract is way too long and went beyond the limit of the journal's guideline of 200 words max.

Methods

Line 160 - This explanation is a study limitation and should be appearing in Discussion, not in methods. 

Recheck Fig 1 caption. 
